# Interaction between Fexofenadine and CYP Phenotyping Probe Drugs in Geneva Cocktail

**DOI:** 10.3390/jpm9040045

**Published:** 2019-10-02

**Authors:** Marija Bosilkovska, Gaelle Magliocco, Jules Desmeules, Caroline Samer, Youssef Daali

**Affiliations:** 1Division of Clinical Pharmacology and Toxicology, Geneva University Hospitals, 1205 Geneva, Switzerland; marija.bosilkovska@hcuge.ch (M.B.); gaelle.magliocco@unige.ch (G.M.); jules.desmeules@hcuge.ch (J.D.); caroline.samer@hcuge.ch (C.S.); 2Faculty of Medicine, University of Geneva, 1205 Geneva, Switzerland

**Keywords:** precision medicine, P-glycoprotein, phenotyping, cocktail

## Abstract

Drug metabolic enzymes and transporters are responsible for an important variability in drug disposition. The cocktail approach is a sound strategy for the simultaneous evaluation of several enzyme and transporter activities for a personalized dosage of medications. Recently, we have demonstrated the reliability of the Geneva cocktail, combining the use of dried blood spots (DBS) and reduced dose of phenotyping drugs for the evaluation of the activity of six cytochromes and P-glycoprotein (P-gp). As part of a study evaluating potential drug–drug interactions between probe drugs of the Geneva cocktail, the present paper focuses on the impact of cytochromes (CYP) probe drugs on the disposition of fexofenadine, a P-gp test drug. In a randomized four-way Latin-square crossover study, 30 healthy volunteers (15 men and 15 women) received caffeine 50 mg, bupropion 20 mg, flurbiprofen 10 mg, omeprazole 10 mg, dextromethorphan 10 mg, midazolam 1 mg, and fexofenadine 25 mg alone (or as part of a previously validated combination) and all together (Geneva cocktail). The determination of drug concentrations was performed in DBS samples and pharmacokinetic parameters were calculated. Fexofenadine AUC_0–8 h_ and C_max_ decreased by 43% (geometric mean ratio: 0.57; CI 90: 0.50–0.65; *p* < 0.001) and 49% (geometric mean ratio: 0.51; CI 90: 0.44–0.59; *p* < 0.001), respectively, when fexofenadine was administered as part of the Geneva cocktail in comparison to fexofenadine alone. Consequently, the apparent oral clearance (Cl/F) increased 1.7-fold (CI 90: 1.49–1.93; *p* < 0.001). There was no interaction between the remaining probes. In conclusion, an unexpected interaction occurred between fexofenadine and one or several of the following substances: caffeine, bupropion, flurbiprofen, omeprazole, dextromethorphan, and midazolam. Further studies are necessary to elucidate the mechanism of this interaction.

## 1. Introduction

Variations in the function of cytochrome P450 (CYP) enzymes as a result of genetic polymorphisms or environmental factors such as dietary components, toxins, or drugs can result in interindividual variability in drug response and pharmacokinetics [1]. Thus, the in vivo evaluation of the activity of these enzymes (phenotyping) is of great importance. A cocktail approach involving the administration of multiple CYP specific probe drugs can be used to simultaneously assess their activity. Numerous cocktails for CYP phenotyping have been described in the past decades and used for the identification of factors which influence CYPs activities or for the study of drug–drug interactions during drug development [2,3,4,5].

In addition to metabolizing enzymes, the importance of drug influx and efflux transporters as a source of pharmacokinetic variability in drug response has been underlined recently [6].

P-glycoprotein (P-gp) is one of the best studied and most significant efflux drug transporters. It is expressed in multiple key organs in drug disposition such as the small intestine, blood-brain barrier, kidneys, and liver and, as such, has been involved in a large number of clinically significant drug–drug interactions [7]. As evidence of the importance of this transporter in drug disposition grows, so does interest in assessing its activity. The investigation of the impact of new drugs on in vivo P-gp activity is recommended by both the FDA and European Medicines Agency (EMA) [8,9]. To this aim, several probe drugs such as digoxin, fexofenadine, talinolol, quinidine, and others have been proposed. Their advantages and limitations have been previously reviewed [10]. In its ‘Guideline on the Investigation of Drug Interactions’, the EMA has suggested the use of renal clearance of digoxin for the determination of renal P-gp function. The function of intestinal P-gp was proposed to be assessed in in vivo studies with probes having a lower oral bioavailability, such as fexofenadine (if no OATP1A1 or 1B3 inhibition is expected) or dabigatran [8].

Previously developed cocktails did not include probes for P-gp activity assessment. Therefore, in vivo interaction studies evaluating the impact of a new drug on CYP and P-gp activity should be conducted on separate occasions [11]. To palliate this, we have recently developed the Geneva cocktail composed of caffeine, bupropion, flurbiprofen, omeprazole, dextromethorphan, midazolam, and fexofenadine as probes for the simultaneous phenotyping of CYP1A2, 2B6, 2C9, 2C19, 2D6, 3A, and P-gp, respectively [12]. In a previous study, we showed that the modulation of CYP and P-gp activity in the presence of inhibitors/inducers could be reliably predicted using this cocktail [12].

As part of a study evaluating potential drug–drug interactions between probe drugs of the Geneva cocktail, the present manuscript focuses on the pharmacokinetic interaction between CYP probe drugs and fexofenadine (P-gp substrate).

## 2. Methods

### 2.1. Subjects

Characteristics of the study volunteers were given in more detail in the previous paper devoted to the analytical development of the Geneva cocktail [12]. Briefly, thirty healthy non-smoking Caucasian volunteers (15 women, 15 men) were included in the study. The median age was 23.5 years (range, 18–36) and median BMI was 21.7 (range, 18.4–27.7). All subjects had normal results on physical examination and liver-function tests and were not taking any medications influencing the function of metabolic enzymes or transporters. Six women had oral contraception and 2 had a hormonal implant. Pregnancy was tested using urinary tests, and all urine analyses were negative at the inclusion and at the morning of each study session. Study sessions for women using oral contraception were scheduled outside the one-week break period for monthly oral contraception. Grapefruit juice was forbidden for at least one week before and during the study, and alcohol and caffeine-containing products were restricted at least 48 h before each study session.

This study (registration NCT02391688) was approved by the Ethics Committee of Geneva University Hospitals (ID: 14-061) and the Swiss Agency for Therapeutic Products (Swissmedic). All volunteers were asked to give written informed consent before starting the study.

### 2.2. Study Design

As this work is part of the previously published study evaluating potential drug–drug interactions between probe drugs of the Geneva cocktail, the design was described in more detail in the previous paper [12]. Briefly, it was a randomized, open-label, four-way Latin-square, cross-over study. The four treatment arms were composed of A: caffeine 50 mg, flurbiprofen 10 mg, omeprazole 10 mg, dextromethorphan 10 mg, and midazolam 1 mg; B: fexofenadine 25 mg; C: bupropion 20 mg; and D: the seven-drug cocktail regimen of caffeine 50 mg, bupropion 20 mg, flurbiprofen 10 mg, omeprazole 10 mg, dextromethorphan 10 mg, midazolam 1 mg, and fexofenadine 25 mg (Geneva cocktail). The Pharmacy of Geneva University Hospitals prepared all the treatments for the study in capsule form except omeprazole 10 mg, where the commercially available tablets (Antramups^®^ 10) were used. The pharmacy is accredited by Swissmedic and works under GMP conditions.

At each session, volunteers received one of the treatments A, B, C or D orally. Capillary blood samples from a small finger prick (BD Microtainer, Contact-Activated Lancet, Plymouth, UK) were collected before (time 0) as well as 0.5, 1, 2, 3, 4, 6, and 8 h after drug administration. Capillary whole blood (10 µL) was collected on a Whatman 903 filter paper card (GmbH, Dassel, Germany) using a volumetric micropipette (Rainin, CA, USA). After 30 min of drying at room temperature, DBS cards were packed in sealable plastic bags and stored at −20 °C. Only capillary whole blood samples were used for fexofenadine pharmacokinetic assessment since we previously demonstrated a high correlation with plasma concentrations.

### 2.3. Analytical Methods

The analytical method used in this study for the determination of fexofenadine, CYP probe drugs, as well as the generated metabolites, consisted of a validated high-performance liquid chromatography-tandem mass spectrometry method (HPLC-MS/MS). Details of the development and the validation of the method were reported elsewhere [13].

### 2.4. Genotyping

Genomic DNA was extracted from whole blood (200 μL) using the QIAamp DNA blood mini kit (QIAGEN, Switzerland). *ABCB1* G2677T/A and C3435T polymorphisms were determined by means of a multiplex PCR with fluorescent probes (Lightmix, TibMolbiol, Berlin, Germany) and melting curve analysis on a LightCycler480 (Roche Diagnostics, Switzerland). For description purposes, volunteers were classified into 3 genotype groups (Table 1): (a)homozygous wild-type carriers 2677GG/3435CC (*n* = 4) and homozygous wild-type carriers at one chromosome and heterozygous for the other ex.2677GG/3435CT or 2677GT/3435CC (*n* = 6).(b)heterozygous carriers at both chromosomes (2677GT/3435CT) (*n* = 10) and homozygous mutated at one chromosome + homozygous wild-type for the other ex. 2677TT/3435CC or 2677GG/3435TT (*n* = 6)(c)homozygous mutated carriers at both chromosomes (2677TT/3435TT) (*n* = 3) and 2677GT/3435TT (*n* = 1)

### 2.5. Statistical Analysis

Pharmacokinetic parameters were estimated by standard noncompartmental methods using WinNonlin version 6.2.1 (Pharsight, Mountain View, CA, USA). The results are presented as mean values (± SD) and geometric mean ratios with 90% CIs. Pharmacokinetic parameters of fexofenadine when administered alone or as part of the cocktail were compared using the nonparametric Wilcoxon signed-rank test. All statistical analyses were performed using SPSS software version 22 (Chicago, IL, USA) and *p*-values of ≤0.05 were considered statistically significant.

## 3. Results

Thirty volunteers participated in the study and successfully completed the four study sessions. None of the subjects reported any side effects after single drug or cocktail administration. Detailed results of drug–drug interactions between the CYP probe substrates contained in the cocktail are described elsewhere [14]. An unexpected interaction occurred between fexofenadine and CYP probe drugs. In fact, in comparison to fexofenadine alone, AUC_0–8 h_ and C_max_ decreased by 43% (geometric mean ratio: 0.57; CI 90: 0.50–0.65; *p* < 0.001) and 49% (geometric mean ratio: 0.51; CI 90: 0.44–0.59; *p* < 0.001), respectively, when fexofenadine was administered as part of the Geneva cocktail. Consequently, the apparent oral clearance (Cl/F) increased 1.7-fold (CI 90: 1.49–1.93; *p* < 0.001). There was no difference in the time to reach the maximum concentration (T_max_) (geometric mean ratio: 1.14; CI 90: 0.87–1.50; *p* = 0.082) and the half-life (t_1/2_) (geometric mean ratio: 1.07; CI 90: 1.00–1.13; *p* = 0.054) when fexofenadine was administered alone or as part of the cocktail (Table 2, Figure 1). The individual AUC_0–8 h_ values of fexofenadine at both sessions are presented in Figure 2. This figure also shows the distribution of *MDR1* genotypes regarding SNPs 2677G > T/A and 3435C > T, which are classified as described in the Methods section (Table 1). The AUC values for these genotypic groups are also shown in Table 3. As expected, the volunteers in group (a), which were homozygous carriers of the wild-type exon-21 2677G > T/A (GG) and exon-26 3435C > T (CC), or heterozygous for one of the exons, had lower AUC values in comparison to volunteers in group (c) with the TT/TT or GT/TT genotype (207.3 ± 44.9 vs. 346.7 ± 37.8 when fexofenadine was administered alone and 113.7 ± 77.5 vs. 179.9 ± 49.6, respectively).

## 4. Discussion

Fexofenadine is a P-glycoprotein substrate that is eliminated unchanged mostly through biliary (80%) and urinary (10%) excretion [15]. Its sensitivity as a P-gp substrate has been demonstrated both in vitro and in vivo in *mdr1* knock-out mice as well as in humans with P-gp inhibitors and inducers [10,16,17]. Therefore, fexofenadine has been proposed as an in vivo P-gp probe by several authors as well as by the International Transporter Consortium [18,19,20]. Furthermore, in comparison to other P-gp probes, fexofenadine has advantages such as a short half-life and a good safety profile. For these reasons, it has been included as a probe for P-gp phenotyping in the Geneva cocktail [12].

Besides its numerous advantages, the main limitation of fexofenadine, as well as the other currently available P-gp probes, is the lack of transporter specificity. Recent studies have shown that in addition to P-gp, other transporters such as multidrug resistance-associated proteins (MRPs), and in particular, organic anion-transporting polypeptides (OATPs), are also involved in fexofenadine disposition [21,22].

In this study, an unexpected interaction occurred between fexofenadine and CYP probe drugs. Few mechanisms can give a potential explanation for the 1.7-fold increase in the apparent oral clearance of fexofenadine. These mechanisms include: (1) increased intestinal efflux due to P-glycoprotein induction, resulting in a decrease in systemic exposure; (2) inhibition of OATP1A2 or OATP2B1, resulting in decreased intestinal uptake; (3) increased hepatic OATP1B1 or OATP1B3 uptake, resulting in increased biliary clearance; or (4) increased hepatic and renal P-glycoprotein-mediated excretion.

Since the probe drugs were administered concomitantly and as a single dose, it is unlikely that P-gp induction will occur. Activation of P-gp has been shown in vitro but has not yet been demonstrated in vivo [23]. Activation of hepatic OATPs (OATP1B1 and OATP1B3) by the non-steroidal anti-inflammatory drugs (NSAIDs) diclofenac, ibuprofen, and naproxen have been observed in vitro [24]. It has been suggested that the allosteric binding of these drugs could increase the transport of some OATP substrates (pravastatin) but not of others (sulfobromophthalein). Further studies should be conducted to verify if flurbiprofen, a NSAID contained in the cocktail, would have this effect on OATPs and on fexofenadine disposition. No direct interaction has been described so far between fexofenadine and any of the other administered probes. However, recently, an unexpected interaction has been observed between bupropion and digoxin, another P-gp probe. Administration of a single oral dose of bupropion (150 mg), 24 h before a single oral dose of digoxin (0.5 mg), resulted in an 80% increase of digoxin clearance [25]. The mechanism of this interaction has been investigated in vitro, and it has been proposed that bupropion increased digoxin renal uptake by activation of renal OATP4C1 [26]. In the present case, this transporter is not known to be involved in fexofenadine disposition, and the fraction of fexofenadine eliminated by the renal route is too low to be the only explanation for the observed interaction. 

An inhibition of intestinal OATP1A2 or OATP2B1 seems to be the most plausible explanation for the observed interaction in this study. However, to our knowledge, none of the co-administered drugs are known to have an inhibitory effect on these transporters. 

## 5. Conclusions

An unexpected drug–drug interaction resulting in increased fexofenadine clearance has been observed between this drug and one or several of the following substances: caffeine, bupropion, flurbiprofen, omeprazole, dextromethorphan, and midazolam. Further studies are required to determine the perpetrator responsible for this drug interaction among the six co-administered CYP probe substrates. The exact mechanism of this interaction also remains to be elucidated. 

Because of the involvement of several transporters in their disposition as well as the occurrence of unpredictable interactions, the utility of fexofenadine as a probe drug for P-gp should be questioned. 

## Figures and Tables

**Figure 1 jpm-09-00045-f001:**
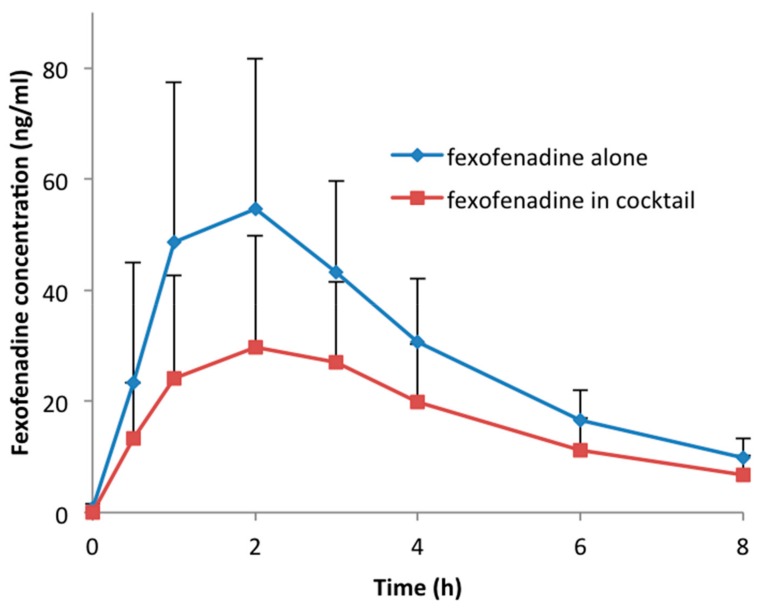
Concentration–time profile of fexofenadine administered alone or within cocktail.

**Figure 2 jpm-09-00045-f002:**
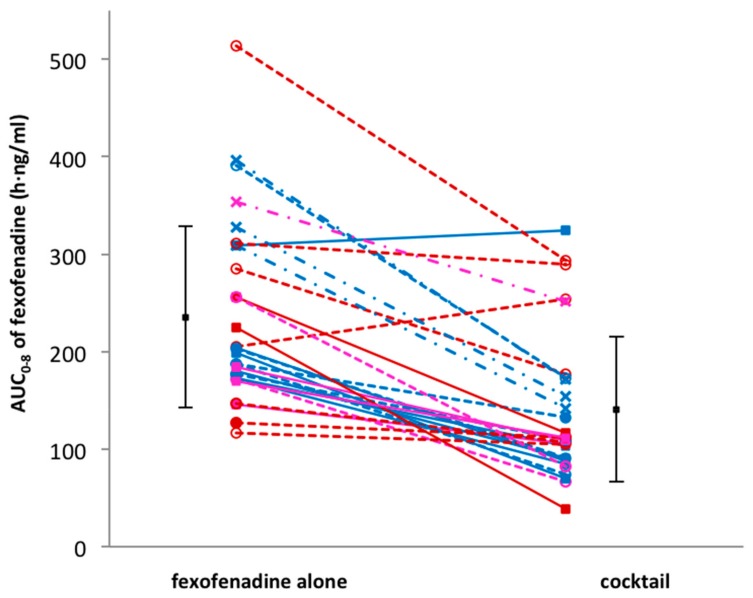
Comparison of individual fexofenadine AUC after administration of fexofenadine alone versus fexofenadine in cocktail. Volunteers in genotype group (a) are shown as solid lines and full squares; genotype group (b) as dotted lines and open symbols; genotype group (c) as dashed lines and cross symbols.

**Table 1 jpm-09-00045-t001:** Stratification of volunteers in genotypic groups according to *MDR1* genotypes regarding single nucleotide polymorphism (SNPs) 2677G > T/A and 3435C > T.

Genotype Group	G2677T/A	C3435T	Number of Subjects
(a)	GG	CC	4
	GG	CT	5
	GT	CC	1
(b)	GT	CT	10
	TT	CC	4
	GG	TT	2
(c)	GT	TT	1
	TT	TT	3

**Table 2 jpm-09-00045-t002:** Pharmacokinetic parameters of fexofenadine administered alone or with cytochromes (CYP) probe substrates.

	Fexofenadine Alone	Fexofenadine in Cocktail	Geometric Mean Ratio	90% CI	*p*-Value
t_1/2_ (h)	2.46 ± 0.30	2.64 ± 0.55	1.07	1.00–1.13	0.054
T_max_ (h)	1.60 ± 0.66	2.01 ± 0.98	1.14	0.87–1.50	0.082
C_max_ (ng/mL)	59.6 ± 27.0	32.9 ± 21.8	0.51	0.44–0.59	<0.001
AUC_0–8_ (h·ng/mL)	235.5 ± 92.8	140.8 ± 74.2	0.57	0.50–0.65	<0.001
AUC_0–∞_ (h·ng/mL)	270.4 ± 102.3	165.7 ± 84.4	0.59	0.52–0.67	<0.001
Cl/F (l/h)	103.9 ± 33.7	189.1 ± 95.1	1.70	1.49–1.93	<0.001

Data are presented as mean values ± SD and geometric mean ratios (fexofenadine in cocktail vs. fexofenadine alone) with 90% CIs. AUC_0–∞_, area under the concentration–time curve extrapolated to infinity; AUC_0–8_, area under the concentration–time curve form 0 to 8 h; CI, confidence interval; Cl/F, apparent clearance; C_max_, maximum blood concentration; t_1/2_, elimination half-life; T_max_, time to maximum blood concentration.

**Table 3 jpm-09-00045-t003:** Stratification of fexofenadine AUC according to *MDR1* genotypes deduced from SNPs 2677G > T/A and 3435C > T.

Genotype Group	AUC_0–8_ Fexofenadine Alone	AUC_0–8_ Fexofenadine in Cocktail
a (n = 10)	207.3 ± 44.9	113.7 ± 77.5
b (n = 16)	225.4 ± 105.9	147.9 ± 74.9
c (n = 4)	346.7 ± 37.8	179.9 ± 49.6

Data are presented as mean values ± SD. For genotype classification see the Methods section. AUC_0–8_, area under the concentration–time curve form 0 to 8 h (last sampling point).

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
