# Peer review of "Interaction between Fexofenadine and CYP Phenotyping Probe Drugs in Geneva Cocktail"

_jpm, 2019, doi:10.3390/jpm9040045_

Round 1

Reviewer 1 Report

Authors investigated the MDR1 gene/drug-fexofenadine interactions and found systemic exposure of fexofenadine decreased and its apparent clearance increased after administration as part of Geneva cocktail in comparison to fexofenadine alone. They postulate that such an unexpected interaction occurred between fexofenadine and one or several of the following substances: caffeine, bupropion, flurbiprofen, omeprazole, dextromethorphan and midazolam. While the mechanistic mechanisms remain unknown, this clinical pharmacokinetic study is well designed and thoroughly undertaken. The unexpected results further demonstrate that there are still significant “unknown known” in this field.

There are a couple of typos which cannot influence the understanding of the scientific merits.

Author Response

Comment 1:

Authors investigated the MDR1 gene/drug-fexofenadine interactions and found systemic exposure of fexofenadine decreased and its apparent clearance increased after administration as part of Geneva cocktail in comparison to fexofenadine alone. They postulate that such an unexpected interaction occurred between fexofenadine and one or several of the following substances: caffeine, bupropion, flurbiprofen, omeprazole, dextromethorphan and midazolam. While the mechanistic mechanisms remain unknown, this clinical pharmacokinetic study is well designed and thoroughly undertaken. The unexpected results further demonstrate that there are still significant “unknown known” in this field.

There are a couple of typos which cannot influence the understanding of the scientific merits.

Answer 1: 

Authors would like to thank the reviewer for the positive comments.

English was revised throughout the manuscript 

Reviewer 2 Report

This well written manuscript uses the technique of measuring pharmacokinetics of a mixture of drugs that cover most common hepatic metabolism pathways with a test drug, in this case fexofenadine, in order to screen more rapidly for significant drug interactions. The mixture  used in this study was unique in that it included a screening drug to detect interactions with the pGp drug transporter. This was the novel aspect of the study. (Geneva Cocktail) The outcome was that the clearance and hence the AUC for fexofenadine was reduced by about 40% by  the cocktail. The authors argue that  this was an unexpected finding and although included in their Cocktail, the result was more like to be an enzyme interaction rather than one with pGp (on the basis of other studies with pGp). They concede that a weakness of the study was the lack of specificity for their transporter probe, and that further studies were needed to determine the the pathway (s) responsible for the interaction. The study demonstrates the use fullness of Cocktail studies as a screening method for drug-drug interactions.

Author Response

Comment 1:

This well written manuscript uses the technique of measuring pharmacokinetics of a mixture of drugs that cover most common hepatic metabolism pathways with a test drug, in this case fexofenadine, in order to screen more rapidly for significant drug interactions. The mixture  used in this study was unique in that it included a screening drug to detect interactions with the pGp drug transporter. This was the novel aspect of the study. (Geneva Cocktail) The outcome was that the clearance and hence the AUC for fexofenadine was reduced by about 40% by  the cocktail. The authors argue that  this was an unexpected finding and although included in their Cocktail, the result was more like to be an enzyme interaction rather than one with pGp (on the basis of other studies with pGp). They concede that a weakness of the study was the lack of specificity for their transporter probe, and that further studies were needed to determine the the pathway (s) responsible for the interaction. The study demonstrates the use fullness of Cocktail studies as a screening method for drug-drug interactions.

Answer 1:

Thank you for your positive comments.

Reviewer 3 Report

Agree the Geneva cocktail is a common approach for evaluating/screening for CYP450 drug interactions.  Intent is that if no changes, can conclude no CYP-mediated drug interactions. However, if an alteration in PK (i.e. clearance, AUC..etc) is observed, then additional experiments are required to delineate the pathway associated with interaction.

This data presented...is only part one...authors need to complete part two then publish.  Information is NOT useful as presented.

In addition, strongly encourage to look at PHASE II metabolism pathways that also might be contributing to alterations in PK.  

Author Response

Comment 1:

Agree the Geneva cocktail is a common approach for evaluating/screening for CYP450 drug interactions.  Intent is that if no changes, can conclude no CYP-mediated drug interactions. However, if an alteration in PK (i.e. clearance, AUC..etc) is observed, then additional experiments are required to delineate the pathway associated with interaction.

This data presented...is only part one...authors need to complete part two then publish.  Information is NOT useful as presented.

In addition, strongly encourage to look at PHASE II metabolism pathways that also might be contributing to alterations in PK.  

Answer 1:

As mentioned in the manuscript, this unexpected drug-drug interaction could be of interest for people using fexofenadine in a cocktail for Pgp activity assessment. It is true that the mechanism behind this interaction is not yet elucidated, and more in-vitro and in-vivo/in-silico studies need to be performed. This work in ongoing in our laboratory and will be published in a separate paper.

Moreover, Since fexofenadine is eliminated unchanged mostly through biliary (80%) and urinary (10%) excretion, the interaction is more likely related to transport inhibition by one of the probes of the cocktail.

Reviewer 4 Report

The manuscript "Interaction between fexofenadine and CYP 2 phenotyping probe drugs in Geneva cocktail" by Daali et al describes the previously unreported finding about fexofenadine and one of the substance present in Geneva cocktail. which consequently increases the oral clearance of fexofenadine .

These are some of my queries and suggestion:

Although this is a very good observation, the exact compound or compounds in the Geneva cocktail with which fexofenadine interacts is not mentioned in the manuscript. Obviously they need more experiments to prove this.

The interaction of fexofenadine with efflux pumps is rather complex and does need more research to confirm their exact mechanism.

The authors have reported a key observation which was not reported before with out any convincing reason as to why they observe such a change in oral clearance.

The language needs to be improved. Also reference 3 needs page numbers.

Author Response

Comment 1:

These are some of my queries and suggestion:

Although this is a very good observation, the exact compound or compounds in the Geneva cocktail with which fexofenadine interacts is not mentioned in the manuscript. Obviously they need more experiments to prove this.

The interaction of fexofenadine with efflux pumps is rather complex and does need more research to confirm their exact mechanism.

The authors have reported a key observation which was not reported before with out any convincing reason as to why they observe such a change in oral clearance.

The language needs to be improved. Also reference 3 needs page numbers.

Answer 1:

Authors would like to thank the reviewer for the positive and insightful comments.

As mentioned in the manuscript, this unexpected drug-drug interaction could be of interest for people using fexofenadine in a cocktail for Pgp activity assessment. It is true that the mechanism behind this interaction is not yet elucidated, and more in-vitro and in-vivo/in-silico studies need to be performed. This work in ongoing in our laboratory and will be published in a separate paper.

Moreover, Since fexofenadine is eliminated unchanged mostly through biliary (80%) and urinary (10%) excretion, the interaction is more likely related to transport inhibition by one of the probes of the cocktail.

The language was checked by a native English speaking scientist

Reference 3 was corrected